# Lung Microbiome in Critically Ill Patients

**DOI:** 10.3390/life12010007

**Published:** 2021-12-21

**Authors:** Mélanie Fromentin, Jean-Damien Ricard, Damien Roux

**Affiliations:** 1Anesthesiology and Intensive Care Department, AP-HP, Hôpital Cochin, 75014 Paris, France; melanie.fromentin@aphp.fr; 2UMR1137 IAME, Université de Paris, INSERM, 75018 Paris, France; jean-damien.ricard@aphp.fr; 3Médecine Intensive Réanimation, AP-HP, Hôpital Louis Mourier, DMU ESPRIT, 92700 Colombes, France; 4Institut Necker-Enfants Malades, Université de Paris, INSERM U1151, CNRS UMR 8253, 75015 Paris, France

**Keywords:** lung microbiome, 16S rRNA gene, high-throughput sequencing, dysbiosis, metagenomics, ventilator-associated pneumonia, mechanical ventilation, acute respiratory distress syndrome, lung virome, lung mycobiota

## Abstract

The historical hypothesis of sterility of the lungs was invalidated over a decade ago when studies demonstrated the existence of sparse but very diverse bacterial populations in the normal lung and the association between pulmonary dysbiosis and chronic respiratory diseases. Under mechanical ventilation, dysbiosis occurs rapidly with a gradual decline in diversity over time and the progressive predominance of a bacterial pathogen (mainly Proteobacteria) when lung infection occurs. During acute respiratory distress syndrome, an enrichment in bacteria of intestinal origin, mainly Enterobacteriaceae, is observed. However, the role of this dysbiosis in the pathogenesis of ventilator-associated pneumonia and acute respiratory distress syndrome is not yet fully understood. The lack of exploration of other microbial populations, viruses (eukaryotes and prokaryotes) and fungi is a key issue. Further analysis of the interaction between these microbial kingdoms and a better understanding of the host−microbiome interaction are necessary to fully elucidate the role of the microbiome in the pathogenicity of acute diseases. The validation of a consensual and robust methodology in order to make the comparison of the different studies relevant is also required. Filling these different gaps should help develop preventive and therapeutic strategies for both acute respiratory distress syndrome and ventilator-associated pneumonia.

## 1. Introduction

The lung microbiome has been studied for a decade. The key result of the earlier studies was to contradict the hypothesis of sterility of the lung, a notion long taught to medical students [1,2]. Later studies mostly focused on chronic pulmonary diseases such as COPD, asthma or cystic fibrosis [2,3]. In the present review, we chose to focus on the acute setting of patients hospitalized in intensive care units (ICU). First, we will present a critical review of the data regarding the lung microbiome in ventilated patients. Second, we will tackle the definition of the lung microbiome and how it should be studied. A short discussion will also evocate the methodological aspects that may limit the comparability of studies. Finally, we will present some perspectives on the topic of ventilator-associated pneumonia, the most frequent nosocomial infection in ICUs.

## 2. Lung Microbiome in Critically Ill Patients

The microbiome is a diverse ecosystem that includes all host-associated microorganisms and their genomes [4]. These microorganisms belong to various kingdoms including some potential pathogens such as bacteria, viruses and fungi. Almost all studies on the “lung microbiome” provided a limited view of these populations since they focused on only part of the kingdoms, especially the bacterial microbiota.

### 2.1. Lung Bacterial Microbiota

#### 2.1.1. Lung Bacterial Microbiota and Invasive Mechanical Ventilation

Studies to date have been mostly descriptive. A first work demonstrated in 2007 the considerable diversity of microbial populations in bronchial aspirates collected from ventilated patients colonized with *P. aeruginosa* [5]. Since high-throughput sequencing was not gold standard, this very first study used 16S-rRNA clone libraries (PCR amplification, cloning into a vector and sequencing). In 2012, based on a similar methodology for bacterial identification, Bousbia et al. also observed a high bacterial diversity in bronchoalveolar lavage (BAL) from ICU patients mostly ventilated for community-acquired pneumonia [6]. A large repertoire of 146 bacterial species belonging to seven phyla was identified, of which 73 bacterial species had never been described in infected lungs. Subsequently, most studies used high-throughput sequencing of 16S-rDNA hypervariable sequences to explore the lung microbiota. Smith et al. studied the microbiota of 15 uninfected ventilated patients admitted to a surgical unit whose BAL was negative in conventional culture [7]. The same phyla were identified in BAL using sequencing of the V4 hypervariable region of 16S-rRNA genes with an Ion Torrent^®^ sequencer. Most patients had profiles with a high degree of alpha diversity, and inter-individual variation was mostly apparent at the genus level (species diversity within a sample from a given individual). These data were snapshots at a given time point, and the question of how the respiratory microbiota changes under mechanical ventilation overtime, likely the most relevant element, has been addressed in more recent works.

In 2016, Kelly et al. described the airway microbiota using Illumina^®^ MiSeq sequencing of the V1-V2 hypervariable region of 16S-rRNA genes in oropharyngeal swabs and endotracheal aspirates. In this work, alpha diversity decreased rapidly after intubation (i.e., decreasing number of bacterial species in the successive samples), followed by a gradual decrease with prolongation of mechanical ventilation in the absence of pneumonia, as compared to unventilated healthy subjects [8]. The microbiota profiles also showed a higher diversity between individuals in the patient group than between controls (beta diversity) due to a tendency for a single-bacterial species to dominate in ventilated patients. The same observation was made in 35 ventilated patients, for whom 111 tracheal aspirates were available. In this population, based on 16S-rRNA gene sequencing on a 454 platform, alpha diversity decreased over time under mechanical ventilation without being influenced by antibiotic therapies [9].

#### 2.1.2. Lung Bacterial Microbiota and Acute Respiratory Distress Syndrome

Beyond the specific effect of mechanical ventilation on the lung microbiota, acute respiratory distress syndrome (ARDS) or severe systemic inflammatory response syndrome (SIRS) may have an impact on its composition, directly or by enrichment from the gut microbiome [4]. Only a few studies have explored these aspects in critically ill patients. However, the relationship between the gut and the lung microbiome has been well described in asthma or cystic fibrosis and is referred to as the “gut−lung” axis [3,10].

In BAL from 68 patients with ARDS, using V4 region MiSeq Illumina sequencing, *Bacteroides* species were observed in 33% of cases, as compared to only 3% in those from healthy controls [11]. The same authors found, in BAL from 91 mechanically ventilated patients, that the presence of bacteria from the gut microbiome in the lung microbiome was associated with the presence of ARDS [12]. They thus suggested a potential common mechanism, as yet undetermined, explaining the role of the gut microbiome in these pathologies. Similarly, Panzer et al. explored the lung microbiota using MiSeq Illumina sequencing of the V4 region of the 16S-rRAgene in endotracheal aspirates of critically ill trauma patients. In this population, the subsequent development of ARDS was related to the composition of the pulmonary microbiota at 48 h, characterized by an enrichment in *Enterobacteriaceae* and in certain specific taxa such as *Prevotella* and *Fusobacterium* also predominant in the lung microbiome of smokers at baseline (*p* = 0.04 in PERMANOVA) [13].

Two other studies explored the lung microbiota in BAL of ARDS patients [14,15]. Kyo et al. used an Ion One Touch platform to sequence the V5-V6 hypervariable regions of the 16s-rDNA, whereas Schmitt et al. used an Illumina MiSeq sequencing of the V4 region. In both studies, there was a decrease in alpha diversity in ARDS patients as compared to non-ARDS ventilated patient controls. However, the high heterogeneity of the lung microbiota in ARDS patients did not allow individualization of a specific profile [15]. Table 1 summarizes the results of the different comparative studies. Further studies, with comparable methodologies, are needed to better characterize the role of the different actors in the vicious circle between dysbiosis, inflammation and lung injury, and to determine the role of enrichment of the lung microbiota with bacteria from the gut microbiota.

#### 2.1.3. Bacterial Microbiota and Lung Infections

The bacterial lung microbiota has not been extensively studied in the context of acute lung infections, in particular under mechanical ventilation. Flanagan et al. were the first in 2007 to clone and sequence r16S DNA from bronchial aspirates and BAL of mechanically ventilated ICU patients who were colonized with *P. aeruginosa* [5]. Identified bacteria belonged mainly to the three major phyla previously described: *Bacteroidetes, Firmicutes* and *Proteobacteria,* and among them the less abundant species belonged to the flora of the oropharyngeal, nasal and gastrointestinal tracts such as *Lactobacillus*, *Enterococcus* and *Veillonella*. During the antibiotic course, a decrease in the diversity of the microbiota was observed along with the significant predominance of *P. aeruginosa* despite its in vitro susceptibility to the administered treatment. From these results, it appears, on the one hand, that the oropharyngeal and digestive microbiota could be an important source of the pulmonary microbiota change during mechanical ventilation, and, on the other hand, that certain non-pathogenic species could have a protective effect against the development of a ventilator-associated pneumonia (VAP). This could act as a commensal barrier flora of which the reduction could be deleterious [4,11].

Interestingly, based on the studies on the respiratory microbiota of ICU ventilated subjects, no specific profile could distinguish acute pneumonia from VAP [6].

More recently, Zakharkina et al. compared the evolution of the respiratory microbiota in ventilated patients who had or had not developed a VAP using 16S-rRNA gene sequencing on a 454 platform [9]. The greater heterogeneity of the bacterial populations in patients who developed a VAP explained the greater increase in beta diversity compared to the ventilated control group. A concept seems to be emerging that the bacteria responsible for VAP (*Staphylococcus*, *Acinetobacter* and *Pseudomonas*) would exclude other bacterial communities [9].

Identification of risk markers within the lung microbiota is probably the most relevant question. Emonet et al. recently attempted to identify metataxonomic risk markers for the occurrence of VAP from the time of intubation to the day of VAP diagnosis using V3-V4 regions MiSeq Illumina sequencing of BAL samples [16]. They did not observe a significant difference in the lung microbiota evolution between patients with VAP and control ventilated patients at any time point. However, tracheal aspirates from patients with VAP contained more *Gammaproteobacteria* (including notably *Pseudomonas* spp, *Enterobacteriaceae*) three days before VAP diagnosis [16]. In parallel, oropharyngeal swabs from these same patients with VAP contained fewer *Bacilli* (*Enterococcus* spp, *Streptococcus* spp, *Lactobacillus* spp, and *Staphylococcus* spp) on ICU admission. The authors used this difference to classify patients between a VAP group and a control group, with good diagnostic performance. However, their results need to be confirmed in other settings and with a greater number of patients. The results of the main studies concerning mechanically ventilated patients and VAP are summarized in Table 2.

Overall, further studies are needed to deepen and dynamically analyze bacterial communities in order to improve the pathophysiological understanding of VAP development, from intubation to the infectious development through colonization. One of the main objectives would be to identify early biomarkers predictive of VAP development.

### 2.2. Lung Virome

Like bacteria, a large panel of viruses reside in the respiratory tract. *Anelloviridae* and *Redondoviridae* are the most prevalent families of DNA viruses [17]. *Adenoviridae*, *Herpesviridae* and *Papillomaviridae* are also often identified. In addition, bacteriophages are common in the respiratory tract. If their role seems important for the development of innate immunity, the interaction between viruses and the immune system seems also to play an important role in the development of respiratory diseases [17]. Viral colonization and infection, particularly through the respiratory tract, begins at birth and is involved in the development and regulation of the innate and adaptive immune system [18]. Each individual has asymptomatic replication that continuously stimulates the immune system, such as herpes simplex virus infections [19]. At the same time, acute infections such as respiratory syncytial virus (RSV) can lead to immune dysregulation which can persist for months after resolution of the infection [20]. An alteration of the respiratory virome is associated with a variety of chronic respiratory pathologies such as allergies, asthma, COPD or pulmonary fibrosis [21]. Although no causal link can be asserted, these studies suggest a pathophysiological role for viruses in respiratory disease that should be explored in ICU ventilated patients.

It is important to mention that the analysis of the virome is much more complex and expensive than the analysis of the bacterial microbiota [4]. Firstly, there are two main types of viruses: eukaryotic viruses and phages (bacterial viruses). Secondly, the absence of a conserved genomic region throughout the viral kingdom prevents the amplification of a sequence of interest, such as the 16s rDNA for bacteria. This therefore requires in-depth sequencing (shotgun sequencing) after potential enrichment methods or human DNA depletion. To work around this issue, many studies only perform targeted PCR. This obviously prevents the identification of viruses that have not been specifically sought.

#### 2.2.1. Virome and Invasive Mechanical Ventilation

The impact of mechanical ventilation on respiratory virome is still unclear and has not been studied much. However, the human virome seems to be strongly altered during hospitalization in ICUs, in connection with viral reactivations, in particular from the herpes group. The latter could be associated with a prolongation of the length of hospitalization and excess mortality [22,23]. In a population of patients with ARDS or VAP, lung biopsies revealed cytomegalovirus (CMV)-related lung damage in 29 to 50% of subjects [24]. In septic patients, CMV reactivation (17%) as well as viral reactivations of Epstein−Barr Virus (EBV) (48–53%), HSV1 (14–26%), HSV2 and HHV6 (10–24%) have also been described [25,26].

Although the impact of mechanical ventilation and that of inflammation or sepsis on the lung virome is difficult to determine with precision, several elements underline the importance of studying it in order to obtain a more comprehensive view of the lung dysbiosis of ICU patients.

#### 2.2.2. Virome and Pulmonary Infections

##### Virome and Community-Acquired Pneumonia

Eukaryotic Virome and Community-Acquired Pneumonia

Viral infections are a major etiology of acute community-acquired pneumonia [27,28]. The most frequently identified pathogenic viruses, including in ventilated ICU patients, include rhinoviruses and influenza viruses, followed by human metapneumoviruses, parainfluenza viruses, respiratory syncytial virus, coronaviruses and adenoviruses.

At the same time, the presence of viruses, such as influenza virus or rhinovirus, in the airways may favor the occurrence of bacterial infections, possibly through a bacterial lung dysbiosis, and could be associated with significant excess mortality [29,30,31]. The reciprocal mechanism could also occur [32].

Prokaryotic Viruses and Community-Acquired Pneumonia

The interaction between phages and bacteria in the gut microbiota is an example of symbiosis, which may play an important role in controlling bacterial populations [33]. Bacteriophages exert selective pressure on their bacterial hosts and directly influence the human microbiota, notably by infecting dominant bacterial populations more frequently and thus favoring the persistence of less competitive bacterial populations but also by conferring antibiotic resistance genes [34,35].

##### Virome and Ventilator-Associated Pneumonia

The role of viruses in the occurrence of VAP and their impact on patient outcome depends on the viral species [36]. For instance, CMV reactivation was associated with bacterial superinfections [37]. CMV pulmonary reactivation was also associated with increased duration of mechanical ventilation, ICU length of stay and mortality in a population of 93 mechanically ventilated patients with suspected VAP, whereas the effect of HSV replication in the lung is less clear [26]. However, Luyt et al. observed that HSV bronchopneumonitis could develop in a fifth of all ventilated patients. Such viral reactivation was associated with a worse outcome [38].

Viruses of the *Herpesviridae*, *Paramyxoviridae* and *Picornaviridae* families have been identified in all ventilated ICU patients in the pioneer study of Bousbia et al. that have included lung viral analysis (targeted PCR) [6]. In this study, HSV and CMV were the most commonly identified viruses, and CMV was more frequently identified in patients with a pneumonia than in controls. Interestingly, parainfluenza virus-1 was detected in three VAP patients [6].

In a recent study, Fang et al. aimed to evaluate the diagnostic performance of metagenomic next-generation sequencing (mNGS) in patients with VAP. They identified viruses in 30 out of 72 patients, mostly HSV-1 (*n* = 12) and EBV (*n* = 10) followed by torque teno virus (TTV) (*n* = 5) and CMV (*n* = 4) [39]. Metagenomic studies are clearly needed to comprehensively describe the evolution of the lung virome during invasive ventilation in ICUs, and to determine the role and the mechanisms of viral dysbiosis on VAP development.

### 2.3. Lung Mycobiota

The study of dysbiotic mycobiota and its correlation with pulmonary disease is in its infancy, and the lung mycobiota in ICU patients is almost unexplored [40].

Few studies have evaluated this lung mycobiota using high-throughput sequencing [39,41,42]. In healthy individuals, studies revealed many environmental fungi including *Aspergillus* sp., mold (*Penicillium* and *Cladosporium*) and yeasts belonging to the two main phyla *Ascomycota* (*Candida*) and *Basidiomycota* (*Malassezia*) [40,41,43]. In contrast, the respiratory mycobiota of patients with chronic respiratory diseases is characterized by a dysbiosis with a restriction of diversity and a clear predominance of *Candida* species [41,44].

In most environments, an interaction between bacterial and fungal communities exists, and the evolution of one community induces a modification of the other. Airway colonization by certain yeasts, notably the genus *Candida*, has been observed in 25 to 50% of patients after a few days of invasive mechanical ventilation [45,46]. This colonization was statistically associated with the development of bacterial lung infections [45,47]. It is therefore plausible that bacterial−fungal interactions play an important role in the pathophysiology of VAP. In a multicenter study of critically ill immunocompetent patients over a 4-year period, 214 patients (26%) with airway colonization were matched and compared with 214 unexposed patients [45]. Bronchial *Candida* colonization was found to be an independent risk factor for *Pseudomonas* pneumonia (9 vs. 4.8%) with an adjusted odds ratio of 2.22 [1.00; 4.92] (*p* = 0.049). Interestingly, airway colonization with *C. albicans* in a murine model induced a Th1-Th17 immune response that promoted the development of bacterial pneumonia through the inhibition of bacterial phagocytosis by alveolar macrophages [48]. The same team showed in vitro that *C. albicans* impaired ROS production by alveolar macrophages and that this correlated in vivo with an increased prevalence of *P. aeruginosa* pneumonia in rats. In addition, the same fungal colonization promoted *A. baumannii*, *E. coli* and *S. aureus* pneumonia in rats [49,50], and that such colonization was an independent risk factor for *A. baumannii* pneumonia in mechanically ventilated ICU patients [47].

To date, only three studies have looked specifically at the mycobiota of ventilated patients with pneumonia. Bousbia et al. identified 22 fungal species belonging to two phyla (*Ascomycota* and *Basidiomycota*) in 31 patients with pneumonia and 6 controls [6]. Distinct profiles emerged with certain classes found specifically in cases of VAP, in cases of CAP or in controls. These data could therefore lead to the hypothesis that the mycobiota was modified with the occurrence of a pneumonia and that these modifications were different according to the type of pulmonary infection. However, no association between bacteria and fungi could be made [6]. Krause et al. were also interested in the mycobiota of ICU patients and more specifically in the place of *Candida* within this mycobiota [51]. In this work, a restriction of fungal diversity was observed in ventilated patients with pneumonia, with *Candida* representing 75% of the identified species. An antibiotic therapy had no noticeable effect. More recently, Fang et al. studied the diagnostic performance of shotgun sequencing in VAP and found *Candida albicans*, *Aspergillus* and *Candida tropicalis* to be the three most commonly identified genera or species (70% of the total) [39]. Conventional microbiological techniques showed similar results. In this study, there were no more positive samples with high-throughput sequencing than with conventional microbiology.

## 3. Lung Microbiome in Intensive Care Medicine: Limits and Perspectives

### 3.1. Limits in 2021

Most studies of the lung microbiome have until recently been limited to the sole bacterial microbiota, using 16S rDNA genes (genes encoding 16S ribosomal RNA) sequencing. No study has really focused on the evolution of the mycobiota or the virome in ventilated patients, whereas fungal lung colonization and viral reactivation are extensively described in this particular population [26,52]. Definitely, inter-kingdom interplay in the lung microbiota and its interaction with the host likely play a key role in the pathophysiology of VAP and have to be considered. Addressing the dynamic evolution of the whole lung microbiome composition (including bacteria, fungi and viruses) is thus one of the main challenges in acute respiratory medicine to redefine our understanding of VAP pathophysiology.

Such studies will help to uncover how fungi contribute to the healthy equilibrium in the respiratory tract and how lung dysbiosis evolves during Candida spp. bronchial colonization to eventually favor emergence of VAP. In the same way, exploring how reactivation of eukaryotic viruses plays a role in lung dysbiosis and host response would provide a greater understanding of VAP and ARDS pathophysiology and prognosis. In addition, bacteriophages, viruses of the bacteria, represent probably most of the viral population within the human respiratory tract. It therefore appears essential to study simultaneously the bacterial microbiota, the mycobiota and the virome to obtain a comprehensive view of the lung dysbiosis associated with mechanical ventilation, VAP and ARDS [53]. Setting aside the technical issues of such an approach, the cost of virome analysis notably limits its feasibility.

This approach would need serious technical development. The validation of a unique nucleic acid extraction protocol for bacterial, fungal and viral nucleic acid would enable an investigation of the entire microbiome profile and its evolution in ventilated ICU patients for large cohort studies.

Another issue is the absence of reproducibility of clinical studies exploring the lung microbiome, a fact possibly linked to methodological issues. For instance, major differences exist in protocols applied to sequence the 16s rDNA from one study to another (i.e., airway sampling, DNA extraction, amplification, sequencing, bioinformatics analysis). These differences limit the comparison of published data [54]. Technical bias can be present at each step of the analyses [4]. One of the most complex questions relates to the hypervariable (V) region(s) of the 16S rRNA gene being sequenced, in order to accurately identify the bacterial population [54]. The most informative hypervariable 16S rRNA region may also differ from one environment, or organ, to another [55]. In addition, few studies describing the respiratory microbiome were performed on endotracheal aspirate (ETA) or bronchoalveolar lavage (BAL).

### 3.2. Perspectives

#### 3.2.1. Establishing a Framework for Microbiome Research

The concept of lung sterility and of a sole bacterial species responsible for pneumonia has been largely refuted by the emergence of high-throughput sequencing projects. This simple, original conceptual framework has been replaced by a much more complex model in which the lung is inhabited by very diverse microbial populations, *Bacteroidetes* and *Firmicutes* being predominant in the normal lung microbiota [8]. Disequilibrium in these populations, termed dysbiosis, could be the first step towards pneumonia. During mechanical ventilation, the early changes in the lung microbiome are indeed characterized by a progressive decrease in alpha diversity [8,9] and in the relative abundance of the two main phyla *Bacteroidetes* and *Firmicutes*, which are notably composed of non-pathogenic anaerobes [8,15,16]. These commensal bacteria appear to play a key role in the lung’s immune homeostasis. Under certain conditions during mechanical ventilation in ICU, they are gradually replaced by the phylum *Proteobacteria* [55]. This enrichment of the lung microbiota by gut-associated bacteria, called the “gut−lung axis”, has been well characterized in ARDS. It is less clear in patients with VAP without ARDS [16], and the precise mechanism by which bacteria translocate from the digestive tract to the lung is not clearly understood [11,13].

Whether pulmonary dysbiosis is only a marker of a more general dysbiosis due to the severity of the critical illness or whether it plays a pathophysiological role in the occurrence of VAP or the prognosis of ARDS remains to be clarified [52]. Bousbia et al. showed that the lung dysbiosis in case of VAP is characterized by a dominance of *Gammaproteobacteria* including *Enterobacteriaceae* (Bousbia). In the study of Emonet et al., the relative abundance of *Proteobacteria* increased from 25% to 55% between the intubation time and the third day of mechanical ventilation in patients who eventually developed a VAP, and the relative abundance of *Firmicutes* decreased from 40% to 30% in the lungs and the oropharynx [16]. Interestingly, the observed decrease in alpha diversity was similar, whether patients developed VAP or not [9,16].

The hypothesis of a specific pattern of lung dysbiosis during VAP or ARDS must be confirmed, and the direct role of this dysbiosis in the pathophysiology of these conditions remains to be clarified. Recent data suggest that the initial composition of the upper and the lower airway microbiota could have direct effects on an individual’s risk of lung infection. For instance, the predominance in the upper respiratory tract of *Rothia Lactobacillus* and *Streptococcus* increases the risk of pneumonia [56]. In contrast, *Prevotella melaninogenica* in the nasal microbiome reduced the risk of influenza pneumonia due to an increased influenzae-specific IgA antibodies [57].

Further longitudinal metagenomic studies are now needed to fully characterize pulmonary dysbiosis in ventilated patients who have developed a VAP or an ARDS to understand whether pulmonary dysbiosis is a cause, a consequence or both. These studies will have to use standardized methods that will allow their comparability.

#### 3.2.2. Clinical Application

##### Improvement in Diagnostic Accuracy

One of the daily issues intensivists face is the accurate diagnosis of VAP in ventilated patient. Regardless of the type of respiratory specimen, pathogen identification by conventional culture-based microbiology techniques is time-consuming and requires a minimum delay of 24–48 h. However, reducing the diagnostic delay is crucial to limit the overuse and misuse of broad-spectrum antibiotic therapy while avoiding excess mortality in case of inappropriate antibiotic therapy. In addition, ruling out a VAP will help to reduce antibiotic consumption and thus decrease antibiotic resistance in the hospital setting. In this context, “culture-independent” techniques based on molecular biology such as multiplex PCR are emerging, allowing the detection of pathogens and resistance markers in a delay reduced to few hours [58]. However, as recent studies evaluating the diagnostic performance have found an imperfect concordance with culture for the identification of pathogens and resistance mechanisms of about 50–60% and 60–70% [59,60], further studies are clearly needed before routine use. Other techniques based on high-throughput sequencing should be considered and explored in order to optimize the diagnosis of community-acquired pneumonia (CAP) or VAP.

Metagenomic next-generation sequencing (mNGS) provides a less biased approach which allows universal pathogen detection from clinical specimens and can be viewed as an ideal method to detect simultaneously bacteria, virus and fungi [61]. Recently, mNGS has shown to be helpful for difficult diagnosis, in the case of low inoculum or ongoing antibiotic therapy, or to detect rare pathogens [62,63] (Zhang 2019) (Wang 2020).

However, a large application among ventilated patients has not been really performed until recently. Yang et al. had recently completed a proof-of-concept case-control study in ventilated patients. Among nine cases of CAP, five were positive with conventional culture. In the latter, nanopore revealed communities with high abundance of the bacterial species isolated by cultures. In culture-negative cases, a probable bacterial pathogen was identified in only one case [64]. Li et al. studied 32 patients with respiratory failure, among whom nine were immunocompromised. The overall diagnostic agreement (mNGS vs. culture/smear/PCR) was 75.68%, and the sensitivity was 81.48%. In 13 cases, the detection results were positive by mNGS but negative by culture/smear and PCR. Importantly, in 11 cases, changes in treatment strategies were applied after diagnosis of a *Chlamydia*, *Nocardia*, Human adenovirus or *Aspergillosis* pneumonia based on mNGS result [65].

Only one recent study by Fang et al. has tried to assess the diagnostic performance of mNGS using BAL in 72 patients with clinically suspected VAP, compared to conventional culture [39]. For bacterial identification, the sensitivity and the specificity of mNGS were 97.1% (95% CI, 93.2–101.0%) and 42.1% (95 CI, 30.7–53.5%), respectively, whereas the positive predictive value (PPV) and the negative predictive value (NPV) were 60.0% (95% CI, 48.7–71.3%) and 94.1% (95% CI, 88.7–99.6%), respectively. Regarding bacterial detection, 26 of the 38 conventional culture negative samples were positive using the mNGS methods. For fungal detection, the sensitivity and specificity of mNGS were 71.9% (95% CI, 61.5–82.3%) and 77.5% (95% CI, 67.9–87.1%), respectively, and PPV and NPV were also 71.9% (95% CI, 61.5–82.3%) and 77.5% (95% CI, 67.9–87.1%), respectively. A total of 9 out of 40 samples negative with conventional method were found positive for fungi according to mNGS. Unfortunately, despite mNGS diagnosis of 30 viral pneumonia, diagnostic performance of mNGS for viruses [39] has not been assessed.

Of note, these promising results were performed with next-generation specific platform BIGISEQ^→^ platform [66], or Oxford Nanopore^→^ MinION device (Oxford Nanopore Technologies, UK) [64], techniques that are not currently available in every country or not available enough to respond to the clinical demands of ICUs. Moreover, these studies have been performed with different experimental protocols, sequencing platforms and bioinformatic tools. Further larger studies are therefore required with a similar protocol to confirm the usefulness of such techniques for a large panel of microorganisms, including virus.

##### Prevention of Ventilator-Associated Pneumonia

In parallel to the challenges of VAP diagnosis, VAP prevention is of high importance for the management of ICU patients. Obviously, a better understanding of pathophysiological infectious steps can help to define targeted interventions on the bacterial microbiota, the mycobiota and the virome.

Among them, we clearly need to know more on the complex interplay between these actors and the host environment and host response in order to better act upon emerging dysbiosis. Depending on the local immune balance between pro- and anti-inflammatory status, one can imagine that boosting immune response may be preferable to antibiotics that may further aggravate the dysbiosis at some moments and that at others, preventive antibiotics may be a preferable option. To that effect, the effect of aerosolized amikacin on the lung microbiota composition could be an interesting lead. It will be evaluated in an ancillary study of a randomized controlled study assessing the preventive effect of aerosolized amikacin in patients at high risk of VAP [67]. In this study, we will also test the impact of aerosolized amikacin on the gut microbiota in order to confirm the very low antibiotic selective pressure of this therapy (on-going work).

Targeting very specific bacterial strains with bacteriophages may also be an interesting field to treat lung dysbiosis and restore normal flora. The same reasoning may be held with antiviral treatment of viral colonization or co-infection.

## 4. Conclusions

To obtain a comprehensive view of the lung microbiome, including not only bacterial but also viral and fungal data, is of great value to improve our understanding of critical lung illnesses such as VAP or ARDS. The evolution of the lung microbiome over time and the description of its dysbiosis will be key elements to improve diagnosis and preventive measures in ventilated patients.

## Figures and Tables

**Table 1 life-12-00007-t001:** Main comparative studies exploring the lung microbiota in ventilated patients with acute respiratory distress syndrome.

Study	Enrolled Patients	Methods (Sampling and Sequencing)	Main Results
Panzer et al., 2018 [13]	30 ventilated patients (severe blunt traumatism) - 13 ARDS ^1^ patients - 17 non-ARDS patients	ETA ^2^ on admission and 24 h after V4 16s-rRNA MiSeq Illumina sequencer	- Association between ARDS development and lung community composition at 48 h (r2 = 0.08, *p* = 0.04) - ARDS patients: microbiota enriched with *Enterobacteriaceae*, *Prevotella* and *Fusobacterium*
Kyo et al., 2019 [14]	47 ventilated patients: - 40 ARDS - 7 non-ARDS	BAL ^3^ within 24 h after intubation V5-6 16s-rRNA Ion One Touch sequencer	- Decreased alpha diversity in ARDS patient compared to controls (*p* = 0.031) - Copy number of 16S rRNA gene of *Betaproteobacteria* decreased in non-surviving (*n* = 16) vs. surviving patient (*n* = 24). (10^6^ vs. 10^4^; *p* < 0.05)
Dickson et al., 2020 [11]	91 ventilated patients - 17 ARDS - 84 non-ARDS	BAL within 24 h of ICU admission V4 16s-rRNA MiSeq Illumina sequencer	- Increased relative abundance of *Enterobacteriaceae* in ARDS patient (12.5% vs. 0.8%) (*p* = 0.002). - Association between presence of gut associated bacteria in the lung microbiota and the ventilator-free days at day 28 (*p* = 0.003)
Schmitt et al., 2020 [15]	30 ventilated patients (surgical) - 15 patients with sepsis-induced ARDS - 15 controls	BAL at ARDS onset (D0 ^4^, D5 ^5^, D10) V4 16s-rRNA MiSeq Illumina sequencer	- Lower alpha diversity in BAL of ARDS patients vs. controls (Shannon index 3 (2;3.6) vs. 1 (0.5;1.5); *p* = 0.007) - Decrease in anaerobic bacteria *Prevotella* spp (*p* = 0.0033) and *Veillonella* spp (*p* = 0.0002) in ARDS patient - Decreased alpha diversity associated with increased length of mechanical ventilation (ρ = −0.48, *p* = 0.009)

^1^ acute respiratory distress syndrome; ^2^ endotracheal aspirate; ^3^ bronchoalveolar lavage; ^4^ day following intubation, ^5^ five days post-intubation.

**Table 2 life-12-00007-t002:** Main comparative studies exploring the lung microbiota in ventilated patients with ventilator-associated pneumonia.

Study	Enrolled Patients	Methods (Sampling and Sequencing)	Main Results
Kelly et al., 2016 [8]	- 15 MV ^1^ patients from medical intensive care unit - 12 healthy unventilated patients	ETA ^2^ and OS ^3^ within 24 h of orotracheal intubation and every 72 h after V1–V2 16s-rRNA MiSeq Illumina sequencer	- Lower alpha diversity in intubated patients than healthy controls (*p* = 2.3 × 10^−13^) - Decreasing alpha diversity overtime in URT ^4^ of VAP ^5^ patient (*p* = 0.0015) - Higher beta diversity in MV patients than in healthy controls
Zakharkina et al., 2017 [9]	- 11 ventilated patients with VAP ^5^ - 18 ventilated patients without VAP - 6 HAP ^6^/CAP ^7^ - non ventilated control patients	- BAL ^8^ for VAP suspicion - ETA at ICU ^9^ admission and twice a week thereafter 16s-rRNA 454 platform	- Decreased alpha diversity associated with increased length of mechanical ventilation (fixed effect regression coefficient (β): −0.03 CI95% [−0.05; −0.005]) - Increase in β diversity for VAP patients (*p* = 0.03)
Emonet et al. 2019 [16]	- 16 late onset confirmed VAP patient - 38 matched ventilated controls	- ETA and OS at five time points during MV including the diagnosis of VAP (DVAP) and three days later (DVAP +3) V3-V4 16s-rRNA MiSeq Illumina sequencer	- Progressive increase in *Proteobacteria* and decrease in *Firmicutes* (40% vs. 30%) in OS and ETA of VAP patients - Greater initial abundance of the *Bacilli* class in ETA from control patients - Association between presence of gut associated bacteria in the lung microbiota and the ventilator-free days at day 28 (*p* = 0.003)

^1^ mechanically ventilated; ^2^ endotracheal aspirate; ^3^ oropharyngeal swab; ^4^ upper respiratory tract; ^5^ ventilator-associated pneumonia; ^6^ hospital-acquired pneumonia; ^7^ community-acquired pneumonia; ^8^ bronchoalveolar lavage; ^9^ intensive care unit.

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
