# Peer review of "Lung Microbiome in Critically Ill Patients"

_life, 2021, doi:10.3390/life12010007_

Round 1

Reviewer 1 Report

Lung microbiome in critically ill patients – The purpose of the review is to 1) review current data on microbiome in ventilated patients 2) definition of microbiome and how it should be studied, and 3) perspectives on ventilator associated pneumonia

Lung microbiome in critically ill patients:

  • Lung bacterial microbiota:
    • 2007: bronchial aspirates from ventilated patients with P aeruginosa, identified bacteroides, firmicutes and proteobacteria, decreased diversity with antibiotics
    • 2015: high bacterial diversity in BAL of ventilated patients with CAP
    • Smith: 15 uninfected patients in surgical unit, found same phyla as above
    • 2016: alpha diversity decreases rapidly after intubation, and gradual decreased with prolonged mechanical ventilation, higher diversity in patients versus controls
    • Mouse model: enrichment of pulmonary microbiota with intestinal bacteria in sepsi; alteration of lung microbiome after experimental sepsis altered for at least 5 days, then normalized after two weeks
    • BAL from 68 ARDS patients: bacteroides in 33%, only 3% healthy controls; presence of bacteria from gut microbiome was associated with presence of ARDS
  • Lung virome:
    • Limitations include: absence of a conserved genomic region for amplification; requires in depth sequencing
    • ARDS/VAP: lung biopsies showed CMV in 29-50% of subjects
    • Septic patients: EBV, CM, HSV etc. described
  • Lung mycobiota:
    • Airway colonization of primarily candida in 25-50% of intubated patients, associated with development of bacterial lung infections
    • Bronchial candida colonization independent risk factor for pseudomonas
    • Distinct profiles in VAP, CAP or controls
  • Limitations:
    • Limited to using 16S sequencing
    • Paucity of data on mycobioa/virome
    • Technical issues/reproducibility

Overall, this is an interesting and not very well studied topic. This review highlights the paucity of existing literature and comments on the potential future of the field. I have several issues with the review as it is:

Major points:

  1. Given how important methodology (both in terms of sample collection and sequencing/culture methods) is to interpreting the microbiome, I would suggest having a definition and methodology section as the first section. Would talk about 16S v metagenomic approaches, the pros and cons of BAL versus lung tissue microbiome (it is not practical to obtain lung tissue from critically ill patients, but is still an important point to bring up, generally, in this area of research), and the limitations of BALs (what are appropriate controls? How do avoid contamination of reusable scopes/collection devices?) in the critical care setting
  2. In the review section on lung microbiome in critically ill patients, it would be nice to not only report existing data chronologically, but also to mention the methodology used (ie: 16s sequencing on BAL – they were all done like this? Were there technical differences that could have accounted for different findings between studies?
  3. Section 2.1.2, in reference to mouse models of the microbiome – it is difficult to know what to do with mouse microbiome because of the pathogen free existence of experimental mice. I would not include in a review primarily on translational research studies, but maybe can move to the perspectives – about need for better mouse models to study this given the limitations of pathogen free mice
  4. Section 2.1.2: in the BAL ARDS bacteroides paper – how many of the gut microbiome patients had evidence of aspiration? If not mentioned, would mention this as a confounder
  5. Table 1 and 2: would include microbiome investigative methodology (ie: 16s sequencing?) for each study
  6. Section 2.1.3. Zaharkina et al – bacteria responsible for VAP would exclude other bacterial communities; alternatively, it may be hard to overcome a dominant strain with our current methodologies, particularly as the burden of microbiota is much, much lower than a pathogenic bacteria
  7. Section 2.2: given that this is not very well studied, is the pulmonary virome of the “healthy” lung described? If yes, would mention what is a ‘healthy/normal virome’? ie: they are mostly phages? Would also mention colonization versus acute infections versus ‘commensals’
  8. Section 2.2.1: 29-50% of lungs from ARDS/VAP with CMV is shocking! Was this an unusual patient population? And were these CMV+ lungs thought to be contributive of their underlying ARDS (ie: acute infection)? And in septic patients, CMV/EBV/HSV/HHV6 reactivation – is this serologic analysis? If it is, I would state clearly as a matter of interest with further investigation needed specifically in the lungs
  9. Section 2.2: this entire section needs better delineation of virome in terms of that which contributes to acute infection and damage, and that which is not overtly pathogenic, but immunomodulatory. It currently seems like most of the literature describes acute viral infections, and if so, should be stated as a limitation of the field
  10. Section 2.3. It is interesting that there are distinct mycologic profiles in VAP/CAP/controls – would mention what they are
  11. Section 3.2.1. The title “Concept” is vague. Would recommend a more descriptive title such as “Establishing a framework for microbiome research” or something like that. There is a lot of focus in the latter part of the concepts section on VAP. It would be interesting also to think about ARDS. In fact, almost the entirety of the last section is on VAP and diagnosing VAP, and there is little to nothing on ARDS.
  12. Section 3.2.2.1. The paragraphs on the diagnostic accuracy of VAP is a little misleading, as it is focused a lot on identifying primary pathogens, as opposed to, perhaps, understanding how primary pathogens dysregulate the underlying microbiome and lead to differences in outcomes, for example. The diagnosis of pathogenic bacteria is, in my mind, conceptually very different, than dysbiosis caused by the primary pathogen.

Author Response

Major points:

  • Given how important methodology (both in terms of sample collection and sequencing/culture methods) is to interpreting the microbiome, I would suggest having a definition and methodology section as the first section. Would talk about 16S v metagenomic approaches, the pros and cons of BAL versus lung tissue microbiome (it is not practical to obtain lung tissue from critically ill patients, but is still an important point to bring up, generally, in this area of research), and the limitations of BALs (what are appropriate controls? How do avoid contamination of reusable scopes/collection devices?) in the critical care setting

Response: While we would readily welcome an independent section dealing with these methodological aspects, we believe there is, for the time being, insufficient data comparing techniques and methodological approaches.

In addition, we have recently published a review in which some of these aspects are questioned discussed (Fromentin et al, ICM 2021, doi:10.1007/s00134-020-06338-2).

However, to answer the reviewer’s request, we have added, whenever possible, some aspects of methodological issues or information within the manuscript.

  • In the review section on lung microbiome in critically ill patients, it would be nice to not only report existing data chronologically, but also to mention the methodology used (ie: 16s sequencing on BAL – they were all done like this? Were there technical differences that could have accounted for different findings between studies?

Response: We agree with the reviewer. We did add the methodology used for all studies in the review part. Obviously, the heterogeneity may induce a bias for studies comparison. This is to date one of the most important issue of this research field. Homogenization has been done for gut microbiota studies. It has to be also done for lung microbiota in the close future. We added a sentence highlighting this fact (end of paragraph 3.2.1)

  • Section 2.1.2, in reference to mouse models of the microbiome – it is difficult to know what to do with mouse microbiome because of the pathogen free existence of experimental mice. I would not include in a review primarily on translational research studies, but maybe can move to the perspectives – about need for better mouse models to study this given the limitations of pathogen free mice

Response: Absolutely, these animal data have been removed.

  • Section 2.1.2: in the BAL ARDS bacteroides paper – how many of the gut microbiome patients had evidence of aspiration? If not mentioned, would mention this as a confounder

Response: In the paper from Dickson et al, the most common ARDS risk factors were pneumonia (29.4%), sepsis (26.4%) and aspiration (20.6%). Other etiologies included pancreatitis, trauma and transfusion-related lung injury. The enrichment of lung microbiome with a Bacteroides was observed in 33% of cases, as compared to only 3% in those from healthy controls. Among these 33% ARDS patients, there was no information regarding hypothetical aspiration. Interestingly, since the association between Bacteroides persisted after exclusion of patients with a condition of pneumonia, it seems to be independent to “macroscopic” aspiration. In fact, the enrichment of lung microbiome with Bacteroides was associated with systemic inflammation but not with alveolar inflammation (a marker of pneumonia).

  • Table 1 and 2: would include microbiome investigative methodology (ie: 16s sequencing?) for each study

Response: We agree with the reviewer, and we have added these data in the two tables.

  • Section 2.1.3. Zaharkina et al – bacteria responsible for VAP would exclude other bacterial communities; alternatively, it may be hard to overcome a dominant strain with our current methodologies, particularly as the burden of microbiota is much, much lower than a pathogenic bacteria

Response: Some papers contradicted this hypothesis. Indeed, it happened that some samples of VAP patients found a discordance between the conventional microbiology (that identified a pathogen) and the microbiota in which this pathogenic bacterium was not dominant (Kitsios et al – Frontiers in Microbiology 2018). However, we agree that, in many cases, the overgrowth of a pathogenic bacterial species explained the increase beta-diversity in the group of VAP patients.

  • Section 2.2: given that this is not very well studied, is the pulmonary virome of the “healthy” lung described? If yes, would mention what is a ‘healthy/normal virome’? ie: they are mostly phages? Would also mention colonization versus acute infections versus ‘commensals’

Response: Only few studies tried to describe the healthy human lung virome, notably from sputum or bronchoalveolar lavage of donor lungs (right before lung transplantation). The healthy human lung can be populated a large set of viral communities. (Abbas  et al. Am. J. Transplant. 2017;17:1313–1324 ; Abbas et al. Am. J. Transplant. 2019;19:1086–1097 ; Young et al. Am. J. Transpl. 2015;15:200–209). And the relative abundance of eukaryotic viruses in the respiratory tract virome was highly variable between studies due to the heterogeneity of the methods. (Rascovan et al Annu. Rev. Microbiol. 2016. 70:125–41). Anelloviridae were described as the most prevalent DNA viruses family followed by Redondoviridae and other eukaryotic viruses frequently detected Adenoviridae, Herpesviridae and Papillomaviridae. (Young et al. Am. J. Transpl. 2015;15:200–209). Phage were also identified in the lung microbiome. (Abbas 2019) (Young 2015).

To date, no data are available on the putative effect of phages in the lungs. Further research is required to characterize the lung virome and to determine their role in the pathophysiology of acute respiratory disease among ventilated patients.

We added a sentence at the beginning of the chapter 2.2 to present general data of the normal lung virome.

  • Section 2.2.1: 29-50% of lungs from ARDS/VAP with CMV is shocking! Was this an unusual patient population? And were these CMV+ lungs thought to be contributive of their underlying ARDS (ie: acute infection)? And in septic patients, CMV/EBV/HSV/HHV6 reactivation – is this serologic analysis? If it is, I would state clearly as a matter of interest with further investigation needed specifically in the lungs

Response: These are usual data in this field, obtained from PCR (not serology). Many viruses do replicate in lungs of critically ill patients. However, their pathogenicity is difficult to establish. To date, treatment against CMV or HSV was not associated with a clinical improvement, and is not recommended.

  • Section 2.2: this entire section needs better delineation of virome in terms of that which contributes to acute infection and damage, and that which is not overtly pathogenic, but immunomodulatory. It currently seems like most of the literature describes acute viral infections, and if so, should be stated as a limitation of the field

Response: We agree that, especially in community-acquired pneumonia, studies on airway viruses focused on viral infection or coinfection. In critically-ill patients, during mechanical ventilation, reactivation of many viruses (mainly from the herpes family) is frequently observed. However, acute infection is really hard to define in a population with pre-established acute respiratory distress. Many authors try to discriminate between pathogenic and non-pathogenic situations. But to date, there is no guidelines or easy way to classify individual patient situations regarding viral reactivation in the ICU setting.

  • Section 2.3. It is interesting that there are distinct mycologic profiles in VAP/CAP/controls – would mention what they are

Response: Fungal microbiota obtained from patients irrespective of the type of pneumonia, showed the presence of 22 different species belonging to 2 phyla among which 6 phylotypes had not been previously identified in BAL fluids from pneumonia. Differences in fungal microbiota composition between different type of pneumonia exist at the level of class. Members of Saccharomycetes represented by Candida species were ubiquitously identified in all cohorts and are the only class identify in aspiration pneumonia. Eurotiomycetes, which are represented by Aspergillus, Penicillium and Cladophialophora genera, were dominant in the CAP cohort, whereas Agaricomycetes and an unclassified Ascomycota were only identified in the VAP cohort.

At species level, Candida species were the most common fungal species identified. Environmental fungi, which usually colonize water, food debris and humid building surfaces, were more notably identified in our study than in previous pneumonia studies. Seventeen fungi are identified in only one specific pneumonia cohort (5 in CAP cohort, 6 in VAP cohort, 4 in non-ventilated ICU pneumonia cohort 9 in aspiration pneumonia) suggesting specific mycobiome pneumonia profile.

Since it comes from a single set of data at a single time point, and because these data have not been reproduced so far, we believe it could be misleading to detail such precisely. That is why we just wanted to introduce the concept of a mycobiome “signature” of different types of lung infections.

  • Section 3.2.1. The title “Concept” is vague. Would recommend a more descriptive title such as “Establishing a framework for microbiome research” or something like that. There is a lot of focus in the latter part of the concepts section on VAP. It would be interesting also to think about ARDS. In fact, almost the entirety of the last section is on VAP and diagnosing VAP, and there is little to nothing on ARDS.

Response: We totally agree with the reviewer. The title and the entire paragraph have been modified.

  • Section 3.2.2.1. The paragraphs on the diagnostic accuracy of VAP is a little misleading, as it is focused a lot on identifying primary pathogens, as opposed to, perhaps, understanding how primary pathogens dysregulate the underlying microbiome and lead to differences in outcomes, for example. The diagnosis of pathogenic bacteria is, in my mind, conceptually very different, than dysbiosis caused by the primary pathogen.

Response: While we might agree with the reviewer in the future, to date, identifying a pathogen in order to treat it is the actual conceptual framework of VAP and its management. Data on lung dysbiosis and its relationship with VAP is too scarce, and mostly descriptive to able to suggest an exhaustive and accurate new framework. 

We deeply thank the reviewer who helped us to improve significantly the manuscript.

Reviewer 2 Report

In this review, the authors aimed to summarize the reports of the lung microbiome in critically-ill patients. It is an interesting approach to disprove that the lung is not a sterile organ. The authors have summarized the studies associated with lung microbiota (bacteria, virome, mycobiota) and invasive mechanical ventilation, acute respiratory distress syndrome, and other pulmonary infections. The limitations and need for improvement in diagnostic accuracy to clinically apply the relevant findings have been well discussed. Neat work!

Author Response

Response: We thank the reviewer for the very positive comments on our work.

Round 2

Reviewer 1 Report

This review would benefit from grammatical editing, but overall is a nice review that summarizes what is currently known about the microbiome in critically ill patients and highlights future areas for research. 

Author Response

We thank the reviewer for the positive comments on our manuscript. It has been now edited by an english-native.